# Application of Capacitive-Resistive Electric Transfer in Physiotherapeutic Clinical Practice and Sports

**DOI:** 10.3390/ijerph182312446

**Published:** 2021-11-26

**Authors:** Luis De Sousa-De Sousa, Cristina Tebar Sanchez, José Luis Maté-Muñoz, Juan Hernández-Lougedo, Manuel Barba, Maria del Carmen Lozano-Estevan, Manuel Vicente Garnacho-Castaño, Pablo García-Fernández

**Affiliations:** 1Department of Radiology, Rehabilitation and Physiotherapy, Faculty of Nursing, Physiotherapy and Podiatry, Complutense University of Madrid, 28040 Madrid, Spain; luisdeso@ucm.es (L.D.S.-D.S.); jmate03@ucm.es (J.L.M.-M.); pablga25@ucm.es (P.G.-F.); 2Department of Physical Activity and Sports, Faculty of Health Sciences, Alfonso X University, Villanueva de la Cañada, 28691 Madrid, Spain; jhernlou@uax.es (J.H.-L.); mruizbar@uax.es (M.B.); 3Department of Nutrition and Food Science, Faculty of Pharmacy, Complutense University of Madrid, 28040 Madrid, Spain; mlozan16@ucm.es; 4Nursing Department, Campus Sant Joan de Deu, 08034 Barcelona, Spain; manuelvicente.garnacho@sjd.edu.es

**Keywords:** diathermy, physical therapy specialty, sports medicine, physiotherapy, review

## Abstract

Diathermy techniques embody an oscillating electrical current passaging through the body tissues generating therapeutic heat; use of this technique in the physiotherapy field has been introduced recently, and because there is scarce information, the following review is proposed, aiming to explore the available evidence on applying CRET in physiotherapy clinical practice and sports. A systematic search was led through a keyword search on PubMed, MedLine, DialNet, Scopus, PEDro, Web of Science and Clinicaltrials databases. Including randomised controlled trials and quasi-experimental studies, which applied radiofrequency diathermy in sports and physiotherapy fields, without any restrictions on dates, published in Spanish, English, Portuguese or Italian. Data extraction was conducted through the Cochrane data extraction form and presented in tabular format; 30 articles were included for analysis, and assessment of methodological quality was made through the PEDro scale with a “Good/Fair” general quality score. The nature of existing articles does not allow a quantitative analysis. Conclusion: identified fields of applications were musculoskeletal physiotherapy, treatment of pelvic floor and sexual dysfunctions, as well as dermato-functional physiotherapy and sports, evidencing an increase of skin temperature, enhanced skin and muscle blood perfusion, as well as reporting an increase in oxyhaemoglobin. Further research is needed. Prospero registration number: CRD42020215592.

## 1. Introduction

Electrophysical agents, specifically within the radiofrequency spectrum, include capacitive-resistive electric transfer (CRET), which consists of oscillating energy at specific frequencies generating therapeutic heat in body tissues, a technique known as diathermy [1].

In a preliminary search for evidence, we found that this type of therapy has various effects at the cellular level [2,3,4,5]. CRET interferes with the synthesis and mobilisation of fats in the early phases of adipogenesis [2], simultaneously increasing the presence of mesenchymal cells; more concretely, those stem cells present in damaged tissues that are adipose-tissue derived [3]. In vitro evidence shows an enhancement of connective tissue regeneration after periods of intermittent exposure to a 448-kHz electric current, increasing the amount of mesenchymal cells while not compromising this stem cells capacity of Adipo-, Osteo- and Chondro-differentiation, suggesting that CRET electrotherapy could be applied as a complementary therapy in the healing process of a variety of tissular lesions [3]. When delivered at 570 kHz, however, CRET has been shown to have cytostatic and cytotoxic qualities in neuroblastomas [4], additionally demonstrating at this frequency cytostatic effects in hepatocarcinoma, and biomarkers indications of a variation of malignancy and cell differentiation, suggesting CRET treatment may guide hepatocarcinoma cells towards hepatic cell normal state [5]. In addition to these in vitro effects, other physiological effects of applying CRET were observed, such as increased skin surface temperature and at a greater depth, a temporary increase in intramuscular and topical blood flow [6], enhanced execution of functional movements [7], and reduced pain, making it a useful tool for rehabilitation [8,9].

Given that these physiological effects could benefit the field of sports, the use of CRET should be extended to the sports population. Due to the technique’s novelty and its potential use in sports, a highly sensitive search that includes sports populations is justified, not limiting the search to a population with pathological characteristics or injuries. A literature review that includes international recommendations while not excluding factors significantly relevant to the use and proper management of diathermy technology is, therefore, essential. These factors have not been examined in previous reviews [10] and could account for the current lack of evidence.

### Objectives

To explore the available evidence on applying CRET in physiotherapy clinical practice and sports in the following aspects:(A)Areas of application within physiotherapy clinical practice;(B)Areas of application within the sports field;(C)Demonstrated effects of radiofrequency diathermy;(D)Study quality;(E)Identification of gaps in knowledge for future research.

## 2. Methods

This review followed the methodology recommended by Arksey and O’Mally [11] and Levac et al. [12] and was reported according to the Preferred Reporting Items for Systematic Reviews and Meta-Analyses extension for Reviews (PRISMA-ScR), Protocol under Prospero registration number CRD42020215592.

### 2.1. Search Strategy

A systematic search was conducted up to October 2020 in the following databases: PubMed, DialNet, Scopus, PEDro, Web of Science and Clinicaltrials.gov (accessed on 13 October 2021). The snowball method was included, given that the search string was designed to return the most exhaustive results possible. Authors were contacted via email for those articles that were considered incomplete and for studies registered in Clinicaltrials.gov (accessed on 13 October 2021) that had no result records.

After a preliminary search, a second search was conducted with a strategy that included controlled and natural language, revised by a health science librarian, and undertaken in all the databases mentioned above. An example of our search strategy can be found in Table 1.

### 2.2. Article Selection

We included randomised controlled trials and quasi-experimental studies in which radiofrequency diathermy was applied, which were published in English, Spanish, Italian or Portuguese, with no date restriction. Studies registered in clinicaltrials.gov (accessed on 13 October 2021) were included only when the results field was completed. Publications were excluded if the study was performed on animals, corpses, or in vitro.

The eligibility evaluation was performed by employing CADIMA online software vers. 2.2.3 Julius Kühn-Institut, Erwin-Baur-Str. 27, 06484 Quedlinburg, Germany (www.cadima.info (accessed on 13 October 2021)) following the PRISMA flow diagram, with 2 reviewers evaluating the relevance of the studies on the same platform in an independent, standardised and unmasked manner (LDS, CT). CADIMA also enabled a study of interobserver consistency.

### 2.3. Data Analysis

An extraction sheet was adapted from the Cochrane “Data collection form for intervention reviews for RCTs and non-RCTs-template”. One of the researchers extracted the data from the included studies (LDS), while a second researcher reviewed them (CTS); discrepancies were resolved by a third researcher (PGF). To facilitate the presentation of the data in a systematic and synthesised manner, we presented (in a tabular format) the data relevant to our study objectives and simplified its analysis through a textual narrative synthesis.

### 2.4. Study Quality and Bias Assessment

Instead of using the Cochrane tool for data extraction, we employed the PEDro scale for the quality assessment because it provides a higher probability of reliability in the indexed scores and is a sufficiently reliable tool for use in literature reviews in physiotherapy [13]. Due to the review’s nature, we decided not to exclude those studies with “fair” or “poor” quality nor quasi-experimental studies because this is not an initial priority for this type of research [14]. These studies were included to gather the maximum information on this technology, cover the largest number of fields of application, and explore potential lines of research. We sought to reduce publication bias by including articles registered in Clinicaltrials.gov (accessed on 13 October 2021) and reduce result bias by registering the protocol before conducting the review.

## 3. Results

When the search strategy was being carried out, due to the novelty of this technique, it was detected that there could be a nomenclature problem, so it was decided to include a mixture of controlled and natural language allowing a more effective search, resulting in a detailed search string to each database; when analyzing results, we were able to confirm that this nomenclature issue indeed existed. A process was opened with the U.S. National Library of Medicine suggesting indexation of a unified term inside the tree-Number “E02” labelled as “Therapeutics”, more specifically inside “Diathermy” major subject, along with terms such as “Short-Wave Therapy” or “Ultrasonic Therapy”.

Figure 1 shows the PRISMA 2020 flow diagram summarising the selection process. The interobserver consistency (LDS-CTS) showed a “good” level of concordance (kappa index k = 0.704); discrepancies were resolved by a third researcher (PGF).

### 3.1. Included Studies

A total of 30 articles were included, 21 studies assessed CRET usage in the physiotherapeutic clinical practice, 7 covered the demonstrated effects of radiofrequency diathermy in concordance with secondary objectives of the study, leaving the remaining two studies to evaluate the sportive application of CRET. The 30 included articles are summarised in Table 2 and are ordered first by the degree of evidence implied by the study design (randomised, non-randomised and quasi-experimental pre/post clinical trial), followed by the PEDro score. Articles with the same score are ordered by the publishing journal’s impact factor. The interobserver consistency (LDS-CTS) regarding the PEDro score resulted in a “moderate” level of concordance (kappa index k = 0.570); discrepancies generated were resolved by a third researcher (PGF).

Except for one study performed last century, the studies were current, with more than 50% having been published in the past 2 years, denoting significant topicality. A total of 83% were published in Europe, 13% in Asia and 3% in South America. Only 22 studies were evaluated with the PEDro scale, achieving a mean score of 6.13 ± 1.75 points and were, therefore, classified as having “good” validity. Quasi-experimental reports were used to complement the identification of common usage and existing gaps.

ASIS: anterior superior iliac spine; cm: centimeters, °C: degrees centigrade, CRET: capacitive-resistive electric transfer, dm: decimeters, EC: experimental condition, Hz: Hertz, Kg/cm^3^: kilograms per cubic centimeter, Km/h: kilometer per hour, LD: lymphatic drainage, MD: mean difference, M: men, mm: milimeters, mmHg: millimeter of mercury, mmol/L: milimoles per liter, PFE: Pelvic floor exercises, PSWT: Pulsed Shortwave Therapy, pts: points, P: Score confirmed by PEDro database, Q1:first quartile, Q3: third quartile, s: seconds, ST: standard treatment, T: timepoint, TENS: transcutaneous electrical nerve stimulation, umol/L: micromoles per liter, W: women.

### 3.2. Participants, Experimental Conditions, Apparatus and Frequencies

A total of 1057 participants were included in this review, covering healthy individuals (12.9%) and those with disease (87.1%). Some 73.3% of the experimental conditions included only CRET; the rest were applied in combination with another type of technique. The most common comparison group was placebo (46.6% of the included studies).

The most often applied frequency was 448 kHz, and the mean frequency was 561.83 ± 214 kHz (range, 50–1200 kHz). The mean application time was 22.7 ± 16.9 min, with a mean usage time of 14.1 ± 22.2 min for the capacitive electrode and 11.2 ± 8.9 min for the resistive electrode.

### 3.3. Methodological Quality of the Included Studies

Grouped by score; 4.5% of the studies presented a total score of 9 out of 10 in the PEDro scale, 18.2% achieved a score of 8, 31.8% a total of 7 points while 9.1% a score of 6, 13.6% scored 5 and 4 points, respectively, and the remaining 9.2% obtained the lowest score in this review with a total of 3 points.

It is observed that no study scored positively for the item related to internal validity, more specifically about assessing whether they were able to blind the therapist who applies the experimental technique, it is not entirely surprising since great difficulty in blinding the therapist in behavioural and non-pharmacological interventions has been shown [41] demonstrating in physiotherapy that blinding is sometimes impossible [42]. The second least scored item is also related to the internal validity and blinding, the observer or whoever records the results, in a total of 12 articles, this characteristic is not fulfilled or specified, observer blindness is perhaps the most important according to Schulz et al. [43], emphasizing that when not fulfilled in studies with soft variables, risk of verification bias is greater and results may be compromised, occurring to a lesser extent in studies with hard variables.

Two items were missing in 11 of the studies, one referring once again to internal validity and blindness in this case of the participants, mainly affecting subjective variables that participants must fill in and, to a lesser extent, those hard variables [43], and another assessing interpretability, referring to whether the study provides specific measures and variability, allowing assessment of effect size and its variation.

The criteria where more studies coincide was reporting baseline comparison between the groups, thus that the results can be comparable, increasing the reliability in them, as well as describing results of all subjects who received treatment or who were assigned to the control group, reducing the risk of reporting bias.

The external validity item is presented for guidance purposes, and it is for this reason that this criterion is not indexed in the total score.

## 4. Discussion

This review collected and synthesised data from the scientific literature on the use of CRET, the existing evidence of its sportive usage is limited, and the main body of the results covered a wide variety of physiotherapeutic clinical fields, such as application in musculoskeletal pathology, pelvic floor dysfunction, dermato-functional application and gastrointestinal pathology. To a lesser extent, its physiological effects on healthy subjects and standard nomenclature. The growing interest in this technique is evident in the notable increase in such studies in recent years. However, there is an identified need to methodically and critically analyse these studies’ content and quality in a broader manner than the existing review [10].

### 4.1. CRET Effects on Temperature and Irrigation 

Numerous studies have shown CRET-induced changes in skin temperature, which is more effective in the resistive case than the capacitive and more prolonged, being up to 5.3% higher in a study that compared both cases during a follow-up of 45 min [17] and another with similar results in a follow-up of 10 min [6]. Compared with other technologies, CRET has shown differences in increased skin temperature compared with shortwave applications [21]. This temperature increase is not limited to the surface and has been recorded up to 20 mm deep [26]. In another test, these skin temperature changes were recorded up to 164 min later [28]. A higher oxyhaemoglobin concentration had been reported when applying heat through CRET than by surface heat, with a difference of up to 9% and higher ranges (56.3–77.9 vs. 48.2–74.0 µmol/L) [26].

In addition to these effects, increased blood perfusion has been reported both at the skin and muscle levels. At the skin level, a study showed a 6-fold larger effect size between pre-treatment and post-treatment than placebo (−24.8% [interquartile range (IQR) 16.8] vs. −3.97% [IQR 22.01]). At the muscle level, this difference was greater than with placebo (2.06% [IQR 3.3] vs. 0.01% [IQR 0.7]) [6]. More studies are needed to deepen the investigation into the differences between the capacitive-resistive application and the effects of clinical applications. The mean PEDro score in this section was 6.60 ± 1.14.

### 4.2. Applications in the Field of Sports

Although the physiological effects of CRET represent a theoretical improvement in sports performance, its use has only been evidenced in a clinical trial on runners, in which tests were performed at various distances. The trial identified no physiological changes between the groups. After CRET treatment, however, the length, angle, height, and frequency of the stride increased compared with the control group [29]. This study scored 5/10 on the PEDro scale, highlighting the need to perform studies that confirm these results and address its potential in sports performance.

A quasi-experimental pre/post study suggested that this technology presents excellent clinical utility by reducing the number of sessions and treatment duration when CRET is applied to varying degrees of muscle conditions in athletes [35]. Higher-quality randomised clinical trials are needed to confirm these statements.

### 4.3. Applications in Musculoskeletal Conditions

Potential improvements in knee osteoarthritis were assessed in two studies with the Osteoarthritis Index from the Universities of Western Ontario and McMaster. The first study showed no significant intergroup improvements but managed to exceed the minimum clinically detectable difference up to 3 months later [7]. The second study showed a significant difference after treatment [20]. By using a different scale (the Lower Extremity Functional Scale), another study showed the effectiveness of CRET with up to 20 points of difference between the groups after treatment [15]. Significant changes in pain were observed from the end of treatment to 3 months later [7], reaching close to 1 point of difference in the visual analogue scale compared with placebo and control (95% confidence interval 0.1–1.3) [20]. Another study examined changes in neuropathic pain using the Douleur Neuropathique 4 (DN4) questionnaire, scoring a mean of 0.2 ± 0.52 points compared with 3.5 ± 1.83 for the control group [15]. An increase in quadriceps strength was observed from 1 month to 3 months after the treatment [7]. The study by Albornoz-Cabello et al. [15] observed no changes in the joint range were observed between the groups except in flexion, reaching a difference of up to 20° in the range of motion.

The application of CRET for neck pain is inconclusive. Two trials compared the effects of CRET on chronic [16] and myofascial neck pain [19], using the same main variables, and found no significant differences compared with placebo. Pain improved, and the cervical disability index score was lower [16]. Similarly, two quasi-experimental studies highlighted the research avenues to be addressed, casting doubts on a possible improvement in neck pain [37] in terms of the severity of cervicobrachial pain symptoms [33] after applying CRET.

Notarnicola et al. [8] reported differences in the treatment of low back pain when comparing CRET with laser, highlighting a significant improvement in favour of CRET at 1 and 2 months after treatment. Improvements were shown in pain reduction (measured with a visual analogue scale) and reduced physical disability (measured with the Roland-Morris questionnaire and the Oswestry disability index). In contrast to the promising results described above, a study that added the assessment of pressure pain threshold showed no differences between this technique and placebo and reported reaching the minimum difference clinically detectable within the CRET group [30]. Another randomised clinical trial observed no significant differences compared with placebo either in medication intake or on a quality-of-life questionnaire [32]. Due to temporality, the studies mentioned above might have been conducted due to 2 prior quasi-experimental studies that demonstrated the possible benefits of this technology in low back pain disability [39] and in the severity of the disease’s symptoms [38].

Within musculoskeletal applications, a quasi-experimental study recently analysed the possible benefits of CRET in rotator cuff tendinopathy, indicating positive effects on resolving pain and oedema [9], providing hypotheses for future research, which are necessary to improve the study design and methodological quality. The mean PEDro score in this section was 6.2 ± 2.2 points.

### 4.4. Pelvic Floor Applications

One study assessed the possibility that CRET could have beneficial effects for women after childbirth. The study found no significant differences in pain or medication intake. However, there was a reduction in discomfort during sitting, presenting a prevalence within the treatment group of 12% ± 41.1% compared with the placebo group (20% ± 64.5%) [18]. Likewise, childbirth has been shown as one of the possible causes of dyspareunia [44]. A quasi-experimental study suggested that applying CRET reduces intercourse pain and improves maximum/mean muscle strength [40]. This field of application is to be addressed in a more exhaustive study.

Sodré et al. [36] studied the effects of CRET on urinary incontinence after prostatectomy, suggesting benefits for symptom severity and quality of life. Within male sexual disorders (more specifically, Peyronie’s disease), two articles proposed CRET as alternative conservative therapy. In one of the studies, only pain was modified, decreasing in 58% of the participants by up to 2 points on the pain scale [22]. In contrast, the other study showed more encouraging results when combining the therapy with verapamil through hydroelectrophoresis, reducing pain and erectile dysfunction. The most notable results were reported in the combined group, with penile curvature reduced by 9.4° ± 0.76° compared with 12.7° ± 0.77° with diathermy alone [25]. The mean PEDro score in this section was 7.0 ± 1.0.

### 4.5. Applications in Dermato-Functional Physiotherapy

Noites et al. [23] assessed the changes in lipid profiles after CRET compared with placebo and found no significant changes. The authors performed another trial by increasing the number of sessions, showing changes within the therapy group in terms of reduced body mass index, waist circumference and adipose tissue thickness; the decrease in the abdominal fold was significantly less than in the placebo group [24].

For oedematous-fibrosclerotic panniculopathy, a randomised clinical trial revealed changes in weight, perimeters and cellulite classification only when comparing the results within the treatment group [27], with no intergroup changes. Future studies should compare CRET with standard treatment to demonstrate its true potential as a clinical application.

In the case of lymphedema, a clinical trial sought to demonstrate the benefits of CRET compared with pressotherapy and lymphatic drainage but found no significant differences between the study groups [31]. More studies are needed on the diseases addressed, with improved methodological quality. The mean PEDro score in this section was 5.7 ± 1.5.

### 4.6. Other Applications

A quasi-experimental pre/post study in a population with an intellectual disability indicated the possible use of this technique assisted with vacuum therapy to treat chronic constipation, reporting an increase in the number of stools and a reduction in irritability episodes in an evaluation period of 14 days. In improving the stool quality assessed with the Bristol scale [34], the physiological changes at the visceral level suggested by this study represent an opening to a wide field of research regarding possible CRET applications.

The Studies included in this review assess the effectiveness of CRET using a variety of experimental methods, each of them with a series of strengths and limitations. Precipitating conclusions on the efficacy of this technology for a certain clinical issue, delivered in a specific manner, to a defined population and in a particular environment, requires synthesis of a wider spectrum of sources to reach a trustworthy estimation. This process is proposed after the development of the research fields of investigation covered in this study, allowing consequent systematic reviews that address effectiveness as the main research subject. 

Despite the aforementioned reason overall effectiveness of CRET, is suggested from different categories, evidenced mainly in fields of temperature, pain management and biomechanical changes mostly compared over placebo or other techniques such as TENS, Laser or pulsed shortwave therapy, each field of application encouraging further comprehensive research regarding effectiveness.

### 4.7. Nomenclature

A significant discrepancy in terminology was observed between the articles when referring to this type of therapy. Some 16.6% of the articles referred to these interventions as “capacitive-resistive electrical transfer”. The following most repeated terms were “capacitive monopolar radiofrequency-resistive”, “tecartherapy” and “radiofrequency”, each appearing in 13% of the articles. The remaining 45.4% used a combination of terms. There is an urgent need to unify the terms to facilitate their indexing in the health descriptor thesauri, allowing for an adequate level of precision in searches, eliminating synonyms and unifying efforts.

### 4.8. Limitations

The inclusion of the snowball method might have incurred a citation bias. The nature of the articles on this topic does not allow for quantitative analysis of the compendium with its consequent partial loss of objectivity. Lastly, as mentioned before, critical scrutiny of the methodological quality was not completed, given that the objective was to collect as much information as possible on the use of CRET.

## 5. Conclusions

The use of CRET within the field of physiotherapy was evidenced in numerous areas, including musculoskeletal pathology, dermato-functional or pelvic floor treatments, sports injury and the production of positive biomechanical changes in athletes. The proven effects of CRET include increased skin temperature, skin perfusion, muscle perfusion and increased oxyhaemoglobin levels. Evolved studies suggest clinical effectiveness in aspects such as pain management in muscle-skeletal pathology, enhancement of functional movement in knee osteoarthritis and decrease in low back pain physical disability, as well as insinuating improvement of pelvic floor pain management and Peyronie’s disease severity, finally in the dermato-functional field advising a decrease in the abdominal fold when combined with exercise. Numerous lines of future research have been identified, which, in addition to those mentioned above, include the use of CRET in gastric or visceral disease. The overall methodological quality of CRET studies can be considered good to fair, but further research is warranted.

## Figures and Tables

**Figure 1 ijerph-18-12446-f001:**
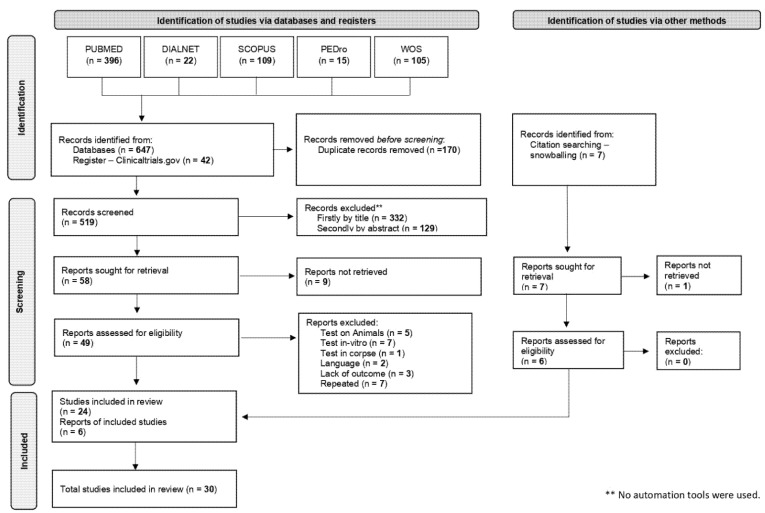
Flowchart showing the article selection process conducted by this review.

**Table 1 ijerph-18-12446-t001:** This table shows the combination of thesaurus and natural terms used on the PUBMED search string and with its corresponding results.

	Conducted	28 October 2020
Search Strategy	Natural, MeSH Terms and Applied Equations	Results Obtained
# 1	Diathermy [MeSH Terms]	
# 2	“Physical therapy modalities” [MeSH Terms] OR “Sports Medicine” [MeSH Terms]	
# 3	“Treatment Outcome” [MeSH Terms]	
# 4	# 1 AND # 2 AND # 3	358
# 5	Wintecare [tiab] OR Winback [tiab] OR Vossman [tiab] OR Indiba [tiab] OR Lavatron [tiab] OR Quilmed [tiab] OR Capenergy [tiab] OR Erbalaser [tiab] OR Tecar [tiab]	23
# 6	Capacitive-Resistive [Title/Abstract]	
# 7	Therapeutics [MeSH Terms] OR “Sports Medicine” [MeSH Terms]	
# 8	# 6 AND # 7	15
	TOTAL, sum of blocks # 4 OR # 5 OR # 8	396

**Table 2 ijerph-18-12446-t002:** Summary of relevant findings for the 30 included articles.

Author, Year, PEDro Score	Participants, Sample Size	Experimental Conditions	Outcome Measures	Measurement Time Points	Main Results
Albornoz-Cabello et al. 2020 [15], 9/10	W (n: 52) and M (n: 32), Adults with patellofemoral pain	EC1: CRET-840 kHz (n: 42) EC2: Control (n: 42)	1. Pain 2. Neuropathic Pain 3. Patellofemoral pain scale 4. lower limb functional scale 5. Range of movement	T0 Baseline T1 After intervention	Pain: lower in EC1 vs. EC2 (6 ± 12.8 vs. 59 ± 16.1 mm). Neuropathic Pain: reduced in EC1 vs. EC2 (0.2 ± 0.52 vs. 3.5 ± 1.83 mm). Patellofemoral pain scale: improved EC1 vs. EC2 (74 ± 13.77 vs. 53 ± 13.49 pts). Lower limb functional scale: improvement in EC1 vs. EC2 (74 ± 14.77 vs. 52 ± 15.84 pts). Range of movement: flexion enhanced in EC1 vs. EC2 (133 ± 7.86 vs. 113 ± 12.58°).
Alguacil-Diego et al. 2019 [16], 8/10	W (n: 14) and M (n: 10) Adults with myofascial chronic neck pain	EC1: CRET-448 kHz (n: 14) EC2: Placebo (n: 10)	1. Pain 2. Neck Disability Index 3. Range of movement	T0 Baseline T1 After first intervention T2 After last intervention	Pain: intragroup in EC1, greater in T0 vs. T1 mean 4.9 cm with a Q1–Q3 dispersion measure of (2.8–6.4) vs. 2.0 cm (1.0–3.0), T0 vs. T2 a mean of 4.9 cm (2.8–6.4) vs. 0.5 cm (0–2.0). Neck Disability Index: in EC1 decrease in T2 vs. T1, 3.0 pts. (2.0–9.0) vs. 10.0 pts. (8.0–12.0) and in EC2 7.0 pts. (6.0–14.5) vs. 14.0 pts. (10.3–20.5).
Kumaran et al. 2015 [17], 8/10	W (n: 9) and M (n: 6), Healthy adults	EC1: CRET-RES448 kHz (n: 15) EC2: CRET-CAP448 kHz (n: 15)	1. Skin temperature	T0 Baseline T1 After intervention T2 45 min After intervention	Higher skin temperature in EC1 (T0–T2) vs. EC2 (T0–T2) (33.2 ± 1.4 vs. 31.5 ± 1.4 °C), within EC1 an increase in T1 vs. T0 (35.0 ± 1.2 vs. 31.1 ± 1.0 °C) and T2 vs. T0 (33.2 ± 1.4 vs. 31.1 ± 1.0 °C); within EC2 an increase in T1 vs. T0 (34.3 ± 1.6 vs. 30.9 ± 1.1 °C) and T2 vs. T0 (33.2 ± 1.4 vs. 30.9 ± 1.1 °C).
Bretelle et al. 2020 [18], 8/10	W (n: 60), Adults with postpartum perineal pain	EC1: CRET-500/300 kHz (n: 29) EC2: Placebo (n: 31)	1. Pain 2. Pain kind 3. Discomfort (Sitting/walking) 4. Paracetamol daily dose	T0 Baseline T1 2 days After intervention T2 30 days After intervention	Discomfort in sitting: decrease in EC1 vs. EC2 (12 ± 41.4 vs. 20 ± 64.5%) when comparing T1.
Coccetta et al. 2019 [7], 8/10 ^P^	W (n: 47) and M (n: 6), Adults with knee osteoarthritis	EC1: CRET-448 kHz (n: 31) EC2: Placebo (n: 22)	1. Osteoarthritis Index 2. Muscle strength scale 3. Pain	T0 Baseline T1 After intervention T2 1 month after intervention T3 3 months after intervention	Osteoarthritis Index: without significant differences, the minimal clinically important difference is exceeded in EC1 vs. EC2 (32.2 vs. 49.4 pts) in T1, (27 vs. 45.9 pts) T2 and (21.4 vs. 43.9 pts) T3. Pain: decrease in EC1 vs. EC2 in T1 (3.4 * vs. 6 mm), T2 (2.8 * vs. 6.4 mm) and T3 (2.6 * vs. 5.9 mm). Quadriceps strength scale: increase in EC1 vs. EC2 in T2 (4.48 * vs. 3.63 pts) and in T3 (4.58 * vs. 3.72 pts) (* significant compared to T0).
Devran et al. 2020 [19], 7/10	W (n: 33) and M (n: 3), Adults with Trapezius myofascial pain	EC1: CRET-N/A + exercise (n: 18) EC2: Placebo + exercise (n: 18)	1. Pain 2. Neck Disability Index 3. Range of movement 4. Trigger points.	T0 Baseline T1 After intervention	No significant differences between experimental conditions.
Kumaran et al. 2019 [20], 7/10 ^P^	W (n: 27) and M (n: 18), Adults with knee osteoarthritis	EC1: CRET-448 kHz + ST (n: 15) EC2: Placebo + ST (n: 15) EC3: ST (n: 15)	1. Pain 2. Osteoarthritis Index 3. Gait evaluation 4. Range of movement 5. physiological variables	T0 Baseline T1 After intervention T2 8 Weeks after intervention T3 3 months after intervention	Pain: an improvement on EC1 vs. EC3, both T0-T1 and T0-T2 (MD 0.82 cm 95% CI 0.32–1.3) and (MD 0.68 cm 95% CI 0.10–1.3) respectively, as well as an improvement in EC1 vs. EC2 (T0-T1) (MD 0.79 cm 95% CI 0.29–1.3). Osteoarthritis Index: improvement in CS1 vs. EC3 (T0-T1) (MD 1.3 pts. 95% CI 0.02–2.6).
Clijsen et al. 2020 [6], 7/10	W (n: 4) and M (n: 6), Healthy adults	EC1: CRET-RES448 kHz (n: 10) EC2: CRET-CAP448 kHz (n: 10) EC3: Placebo (n: 10)	1. Skin perfusion 2. Intramuscular blood flow (distal/proximal) 3. Heart rate 4. Blood pressure 5. Skin temperature	T0 Baseline T1 After intervention T2 2 min after intervention T3 10 min after intervention	Skin perfusion: lower in EC3 vs. EC2 (−24.8 (16.8) vs. −3.97 (22.01) %) in turn, lower in EC2 vs. EC1 (−3.97 (22.01) vs. 23.1 (54.4) %). Distal intramuscular blood flow: greater difference in EC1 (T0-T3) vs. EC3 (T0-T1) (2.06 (3.3) vs. 0.01 (0.7) %) as well as in EC1 (T0-T3) vs. EC2 (T0-T2) (2.06 (3.3) vs. 0.79 (3.3) %), higher skin temperature in EC1 (T0-T3) vs. EC3 (T0-T3) (2.8 (2) vs. −2.3 (1.5) °C). (Average (Q1–Q3)).
Kumaran et al. 2018 [21], 7/10	W (n: 10) and M (n: 7), Healthy adults	EC1: CRET-448 kHz (n: 17) EC2: Placebo (n: 17) EC3: Control (n: 17) CE4: PSWT (n: 15)	1. Skin temperature 2. Skin perfusion 3. Neural conduction velocity	T0 Baseline T1 After intervention T2 20 min after intervention	Skin temperature: significant increase in EC1 vs. the rest of the groups. Skin irrigation: significant increase in EC1 vs. EC2, EC1 vs. EC3, and EC1 vs. CE4. (unspecified results).
Pavone et al. 2017 [22], 7/10	M, Adults with Peyronie’s disease (n: 96)	EC1: CRET-250/500 kHz (n: 64) EC2: Placebo (n: 32)	1. Pain 2. Degree of penile curvature 3. Erectile dysfunction questionnaire	T0 Baseline T1 After intervention T2 3 months after intervention	Pain: improvement in EC1 from T0 to T1 (2 points on the VAS scale in 58% of subjects) (Results not specified).
Noites et al. 2019 [23], 7/10	W, Healthy adults (n:30)	EC1: CRET-480/500 kHz + exercise (n: 15) EC2: Placebo + exercise (n: 15)	1. Lipid profile	T0 Baseline T1 After intervention	Lipid profile without differences between EC1 vs. EC2, changes only in glycerin within EC1 and in EC2 separately, comparing T0 vs. T1 (0.06 ± 0.03 vs. 0.10 ± 0.06 mmol/L and 0.04 ± 0.01 0.07 ± 0.03 mmol/L, respectively).
Vale et al. 2020 [24], 7/10	W, Healthy adults (n: 28)	EC1: CRET-480/500 kHz + exercise (n: 14) EC2: Placebo + exercise (n: 14)	1. Bodyweight 2. Body mass index 3. Waist circumference 4. Subcutaneous adipose tissue thickness 5. Abdominal fold	T0 Baseline T1 After intervention	Body mass index: within EC1 lower in T1 vs. T0 (22.70 ± 2.57 vs. 23.00 ± 2.55 kg/cm^3^). Waist circumference: in EC1 decreases in T1 vs. T0 (78.39 ± 9.52 vs. 80.21 ± 10.21 cm); this difference is greater vs. difference in EC2. Adipose tissue thickness: within EC1 decreases in T1 vs. T0 (12.51 ± 6.40 vs. 10.26 ± 5.44 mm) greater difference vs. difference in EC2. Abdominal fold: decreased in EC1 vs. EC2 (16.32 ± 4.98 vs. 18.01 ± 5 mm).
Maretti et al. 2020 [25], 6/10	M, Adults with Peyronie’s disease (n = 66)	EC1: CRET-500 kHz (n: 32) EC2: CRET-500 kHz + hydro electrophoresis de Verapamil (n: 34)	1. Pain 2. Degree of penile curvature 3. Erectile dysfunction questionnaire	T0 Baseline T1 After intervention T2 3 months after intervention	Pain: Improved EC2 vs. EC1 (0.74 ± 0.11 vs. 1.82 ± 0.11 cm) in T2. Erectile dysfunction questionaire: EC2 improvement vs. EC1 (27.83 ± 0.18 vs. 26.56 ± 0.25 pts) in T2. Degree of penile curvature: Less in EC2 vs. EC1 (9.4 ± 0.76 vs. 12.7 ± 0. 77°). All significant when comparing T2 vs. T0 (intragroup).
Tashiro et al. 2017 [26], 6/10	M, Healthy adults (n: 13)	EC1: CRET-448 kHz (n: 13) EC2: Placebo (n: 13) EC3: Hot-Pack (n: 13)	1. Hemoglobin 2. Temperature at 10/20 mm deep	T0 Baseline T1 periods of 5 min during half an hour	Hemoglobin: increase in EC1 vs. EC2 (98.1 umol/L (79.8–108.9) vs. 92.6 umol/L (78.3–99.5), Mean and ranges), as well as EC1 vs. EC3 (98.1 umol/L (79.8–108.9) vs. 92.3 umol/L (69.8–103.8)), temperature at 20 mm: increase of EC1 vs. EC2 (36.8 °C (35.6–37.5) vs. 35.0 °C (34.0–36.3)), as well as EC1 vs. EC3 (36.8 °C (35.6–37.5) vs. 36.5 °C (34.8–37.1)).
Casa-Almeida et al. 2011 [27], 5/10	W, Adults with edematous fibro sclerotic Panniculopaty (n: 27)	EC1: local CRET-1 MHz (n: 27) EC2: segmented CRET-1 MHz (n: 27)	1. Clinical classification of cellulite severity 2. Perimeters 3. Physiological variables	T0 Baseline T1 After 10th intervention T2 After 20th intervention	Severity classification: no differences between groups, total score improves within EC1 and EC2 when comparing T2 vs. T0 (4.32 ± 2.6 vs. 7.35 ± 2.4 pts and 3.96 ± 2.9 vs. 7.03 ± 2.7 pts, respectively). Perimeter: no differences between groups, within-group EC1 decreases T2 vs. T0 of the iliac crest (75.93 ± 1.1 vs. 77.87 ± 1.1 mm), of ASIS (88.63 ± 1.0 vs. 90.07 ± 1.1 mm) and greater trochanter (96.23 ± 1.2 vs. 97.68 ± 1.2 mm), in EC2 differences in the rest of the perimeters when comparing T2 vs. T0. Weight: decreases in T2 vs. T0 (Results not specified).
Fousekis et al. 2020 [28], 5/10	M, Healthy adults (n: 10)	EC1: CRET-448 kHz (n: 10) EC2: Placebo (n: 10) EC3: CRET-448 kHz + fascia tool (n: 10) CE4: Placebo (n: 10)	1. Skin temperature	T0 Baseline T1 After intervention	Skin temperature: within EC1 and EC3, a significant increase in temperature, Maintenance of this temperature: EC1 vs. EC2 significantly different (39.9 vs. 34.9 °C), EC1 vs. CE4 (39.9 vs. 35.7 °C), likewise EC3 vs. EC2 significantly different (39.7 vs. 34.9 °C) and EC3 vs. CE4 (39.7 vs. 35.7 °C).
Duñabeitia et al. 2018 [29], 5/10	W, Recreational adult runners (n: 14)	EC1: CRET-0.8/1.2 MHz (n: 14) EC2: Control (n: 14)	1. Physiological variables 2. Biomechanical variables (in four tests 10, 12, 14 and 16 km/h)	T0 24 h after excercise T1 72 h after excercise	Biomechanical variables: increase in stride length: in EC1 vs. EC2 (123.39 ± 5.81 vs. 121.91 ± 5.12 cm) at 12 km/h, (139.88 ± 6.47 vs. 138.31 ± 6.57 cm) at 14 km/h and (154.59 ± 8.12 vs. 152.35 ± 9.7 cm) at 16 km/h, stride angle: increase in EC1 vs. EC2 (0.94 ± 0.43 vs. 0.73 ± 0.39°) at 12 km/h, (139.88 ± 6.47 vs. 138.31 ± 6.57°) at 14 km/h (1.91 ± 0.50 vs. 2.01 ± 0.64°) at 16 km/h, stride height: increase in EC1 vs. EC2 (0.90 ± 0.34 vs. 0.85 ± 0.36 cm) at 14 km/h and (1.30 ± 0.38 vs. 1.36 ± 0.44 cm) at 16 km/h, Stride frequency: lower in EC1 vs. EC2 (2.78 ± 0.12 vs. 2.83 ± 0.14 Hz) at 14 km/h and (2.83 ± 0.23 vs. 2.89 ± 0.15 Hz), as well as an increase in these variables when comparing intragroup in EC1.
Rodriguez-Sanz et al. 2017 [30], 4/10	N/A (n:20) Adults with low back and pelvic Pain	EC1: CRET-N/A (n: 10) EC2: Placebo (n: 10)	1. Pain 2. Pain threshold by pressure 3. Physical disability test due to low back pain	T0 Baseline T1 After intervention T2 4 weeks after intervention	There are no significant differences between the groups; in EC1, there is a difference in Pain between T2-T0 greater than the clinically relevant one (2.1 ± 2.1 cm), while in EC2, it is less (0.84 ± 0.7. 2 cm).
Cau et al. 2019 [31], 4/10 ^P^	W, Adults with obesity and lymphedema (n: 48)	EC1: CRET-0.8/1.2 MHz + ST (n:12) EC2: Pressotherapy + ST (n: 12) EC3: LD+ ST (n: 12) CE4: ST (n: 12)	1. Pain 2. Gait evaluation 3. Volume of lower limb	T0 Baseline T1 After intervention	Pain: EC1 improvement in T1 vs. T0 (47.2 ± 24.2 vs. 73.9 ± 21.7 mm). Gait evaluation: decreased EC1 in T1 vs. T0 (13.3 ± 8.1 vs. 21.8 ± 19.7 s). Lower limb volume: decrease in total volume within EC1 in T1 vs. T0 (9.4 ± 2.8 vs. 9.7 ± 2.8 cubic dm) as well as the thigh volume (3.3 ± 1.2 vs. 3.5 ± 1.3 cubic dm), Thigh circumference: decrease in EC1 between T1 vs. T0 (712.8 ± 106.9 vs. 722.35 ± 191.2 mm).
Notarnicola et al. 2017 [8], 4/10	W (n: 43) and M (n: 17), Adults with low back pain	EC1: CRET-450/600 kHz (n: 30) EC2: Laser (n: 30)	1. Pain 2. Disability index 3. Disability questionnaire	T0 Baseline T1 After intervention T2 1 month after intervention T3 2 months after intervention	Pain: decrease in EC1 vs. EC2 (1.6 ± 0.9 vs. 3.5 ± 2.5 mm) in T2 and (0.8 ± 0.7 vs. 4.2 ± 2.4 mm) in T3, disability index: lower in EC1 vs. EC2 (9.4 ± 14.6 vs. 26.8 ± 19.8 pts) in T2 and (6.0 ± 2.7 vs. 26.0 ± 18.6 pts) T3, disability questionnaire: decrease in EC1 vs. EC2 (4.7 ± 2.5 vs. 8.8 ± 6.3 pts) in T1, (2.0 ± 1.9 vs. 9.1 ± 7.0 pts) in T2 and (1.5 ± 1.4 vs. 8.6 ± 6.5 pts) in T3.
Stagi et al. 2008 [32], 3/10	N/A (n: 30) Adults with low back pain	EC1: CRET-485 kHz + massotherapy (n: 15) EC2: Placebo + massotherapy (n: 15)	1. Pain 2. Physical disability test due to low back pain 3. Quality of life questionnaire 4. Medicine intake	T0 Baseline T1 After 4 th intervention T2 After 8 th intervention T3 3 month after intervention T4 6 month after intervention	Pain: difference in its perception and management, improvement within EC1 when comparing T5 vs. T0 (unspecified data). No significant difference between groups.
Sanguedolce et al. 2019 [9], 3/10	W (n: 14) and M (n: 16), Adults with rotator cuff tendinitis	EC1: TENS + iontophoresis + laser + ultrasound + reeducation (n:15) EC2: CRET-N/A + reeducation (n: 15)	1. Pain 2. Specific shoulder test 3. Functional Scale	T0 Baseline T1 4 weeks after intervention T2 8 weeks after intervention	Pain: unspecified statistical contrasts, EC2 vs. CS1 in T1 (5.3 vs. 6.5 cm) and in T2 (3.7 vs. 5 cm).
Takahashi et al. 2000 [33]	W (n: 17) and M (n: 5), Adults with cervical omo-brachial Pain	EC1: CRET-CAP50/651 kHz (n: 22)	1. Severity of symptoms 2. Skin temperature 3. Safety	T0 Baseline T1 After intervention T2 15 min after intervention	Symptom severity: a significant decrease in T1 vs. T0 (6.32 ± 3.36 vs. 9.5 ± 4.75 pts). Skin temperature: significant increase in T2 vs. T0 (30.6–31.3 vs. 29.3–29.8 °C) and T1 vs. T0 (29.7–28.8 vs. 29.3–29.8 °C). Safety: no adverse effect.
Ibánez-Vera et al. 2019 [34]	W (n: 13) and M (n: 6), Constipation in adults with Intellectual Developmental Disorders	EC1: CRET-850 kHz + Vacuum therapy (n: 19)	1. Number of stools 2. Episodes of irritability 3. Scale fecal characteristics	T0 During intervention (14 days) T1 After intervention (14 days)	Number of stools: increase in T1 vs. T0 (10.23 ± 2.32 vs. 8.85 ± 2.34), irritability episodes: decrease in T1 vs. T0 (14 ± 4.26 vs. 15.95 ± 7.05), the scale of characteristics: better in T1 vs. T0 (2.47 ± 0.61 vs. 1.89 ± 0.65 points) significantly.
Ganzit et al. 2000 [35]	W (n: 3) and M (n: 27), Adult athletes with various muscle injuries	EC1: CRET-500 kHz (grade I, II, III muscle injury) (n: 30)	1. Injury resolution by ultrasound image (Days/Sessions)	T0 Baseline T1 After intervention T2 2 weeks after intervention	Injury resolution: by group mean difference, Grade-I: 5.1 sessions in 8 days, Grade-II: 8.6 sessions in 14 days and Grade-III: 11.7 sessions in 19 days.
Sodré et al. 2019 [36]	M, Adults with urinary incontinence after radical prostatectomy (n: 10)	EC1: CRET-0.5/1 MHz (n: 10)	1. Incontinence questionnaire (Short form/Overactive bladder) 2. Loss test 3. Global visual incontinence scale	T0 Baseline T1 After intervention	Incontinence questionnaire: improvement T1 vs. T0 in overactive bladder (6.0 pts. (4.7–8.7) vs. 5.0 pts. (0.7–6.7), mean and standard deviation). Loss test: improvement in T1 vs. T0 (2.0 g (0.0–9.0) vs. 6.5 g (1.7–50.0)), visual incontinence scale: decrease in T1 vs. T0 (4.0 cm (1.7–5.0) vs. 7.0 cm (5.0–8.5)).
Rafaetá et al. 2007 [37]	W (n: 15) and M (n: 5), adults with cervicalgia	EC1: CRET-N/A (n: 20)	1. Pain 2. Cervicalgia questionnaire 3. Visual Scale of improvement	T0 Baseline T1 After intervention T2 2 months after intervention	Pain: Unspecified statistical contrasts, T1 vs. T0 (2.79 vs. 6.63 cm) and in T2 vs. T0 (2.55 vs. 6.63 cm). Questionnaire: without statistical contrast, decreased T1 vs. T0 (16.67 vs. 37.95%) and T2 vs. T0 (12.54 vs. 37.95%).
Takahashi et al. 1999 [38]	W (n: 27) and M (n: 10), Adults with low back pain	EC1: CRET-CAP50/650 kHz (n: 37)	1. Severity of symptoms 2. Skin temperature 3. Safety	T0 Baseline T1 After intervention T2 15 min after intervention	Symptom severity: a significant decrease in T1 vs. T0 (6.2 ± 4.0 vs. 11.5 ± 4.9 pts). Skin temperature: significant increase in T2 vs. T0 (31.1–31.3 vs. 29.2–29.5 °C) and T1 vs. T0 (30.2–30.5 vs. 29.2–29.5 °C). Safety: no adverse effect.
Osti et al. 2014 [39]	W (n: 30) and M (n: 36), Adults with low back pain	EC1: Laser + CRET-450/600 kHz (n: 66)	1. Pain 2. Physical disability test due to low back pain	T0 Baseline T1 8 weeks after intervention	Pain: decreases in T1 vs. T0 (2.63 ± 2.74 vs. 8.1 ± 1.58 cm), disability test: decrease in T1 vs. T0 (23.5 (± 19.8) vs. 53.0 (± 13.0) pts.), even when separating the group by extension of pain, both variables were still significant.
Fernández-Cuadros et al. 2020 [40]	W, Adults with chronic pelvic Pain and dyspareunia (n: 37)	EC1: CRET-448 kHz + Biofeedback + PFE (n: 37)	1. Pain 2. Muscle strength	T0 Baseline T1 After intervention	Pain: decrease in T1 vs. T0 (3.75 ± 2.21 vs. 7.27 ± 1.34 cm), muscle strength: evidence of increased maximum strength in T1 vs. T0 (35.35 ± 20.4 vs. 25.56 ± 15.9 mmHg) as well as the mean force (7.18 ± 4.46 vs. 4.86 ± 3.53 mmHg).

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
