# Peer review of "Application of Capacitive-Resistive Electric Transfer in Physiotherapeutic Clinical Practice and Sports"

_ijerph, 2021, doi:10.3390/ijerph182312446_

Round 1
Reviewer 1 Report
I would like to congratulate the authors for their effort in undertaking this review; however, I do think that it will not qualify to be categorised as a Systematic Review as shown at the top of the paper. The authors have used the term Scoping Review, which is somewhat right.
Regarding the methods, I do have some concerns about the literature search strategy. Emphasis has been given to 'diathermy' and 'capacitive-resistive', which are too specific. The authors need to consider searching more widely using search terms such as 'radiofrequency', 'electromagnetic', 'clinical effects', physiological effects', etc., and then narrow down the results using more specific filter terms. 'Physical therapy modalities' is a less preferred term these days. 'Electrophysical agents' is more widely used. You could also add 'Electrotherapy'. In my view all such terms need to be built into the search process. One more thing, If you are using PubMed for searching, then you do not need to use Medline separately.
The review started with systematic methods, but then it has turned out to be quite descriptive and there is not enough critical element. The results and discussion provides the reader with a summary of what has been published so far; however, it does not provide any information to aid clinical decision making, which is what should be expected from a review. In my opinion the authors should include more specific detail about the methodological quality of the included studies rather than just mentioning the Pedro scores and just making generic conclusive statements about study quality.
Reviewer 2 Report
Thank you to the author for writing an interesting article. However, the article needs significant changes to be published.
Use only one PubMed, MEDline, or Pubmed/Medline
What is the benefit of adding DialNet?
Why not conclude about effectiveness?
Citation missing: “In a preliminary search for evidence, we found that this type of therapy has a variety of effects at the cellular level. ‘
Change interpretation “and concomitantly increases the proliferation of mesenchymal cells; more specifically, mesenchymal stem cells derived from adipose tissue present in damaged tissues.”
Change interpretation, it is an in vitro study “CRET provides enhanced regeneration of connective tissue after periods of intermittent exposure to a 448-kHz frequency, while not compromising the multipotentiality of stem cells in their subsequent adipogenic, chon-46 drogenic, or osteogenic differentiation, and is an effective coadjuvant when dealing with 47 various lesions [3].“
the same situation „and cytostatic effects in hepatocarcinomas, favoring, to some extent, their cell normalisation [5].“
why not knee osteoarthritis? ‘functional movements [7]
Change discussion, make a discussion without results of study.
the conclusion must follow from the study,
Reviewer 3 Report
This systematic review aims to explore the available evidence on the application of CRET in the clinical practice of physiotherapy and sport.
The review is well thought out and methodologically well-conducted. However, there are some issues that the authors should review.
There is no mention of the study's primary objective in the results section, including physiotherapy clinical practice and sports.
The discussion section should begin by summarising the findings on the main objective.
In section 4.5, reference is only made to a study on sports practice. This aspect should be reflected earlier as it is the subject of the study.
Round 2
Reviewer 2 Report
-